# Achievements and Challenges for Higher Education during the COVID-19 Pandemic: A Rapid Review of Media in Africa

**DOI:** 10.3390/ijerph182412888

**Published:** 2021-12-07

**Authors:** Inge K. Sonn, Marieta Du Plessis, Carel D. Jansen Van Vuuren, Janene Marais, Emma Wagener, Nicolette V. Roman

**Affiliations:** 1Centre for Interdisciplinary Studies of Children, Families and Society, University of the Western Cape, Cape Town 7537, South Africa; jmarais@uwc.ac.za (J.M.); nroman@uwc.ac.za (N.V.R.); 2Department of Industrial Psychology, Faculty of Economic and Management Sciences, University of the Western Cape, Cape Town 7537, South Africa; mduplessis@uwc.ac.za (M.D.P.); cdjansenvanvuuren@uwc.ac.za (C.D.J.V.V.); 3Department of Psychology, University of the Western Cape, Cape Town 7537, South Africa; ewagener@uwc.ac.za

**Keywords:** COVID-19 pandemic, global citizenship education, higher education, online education, rapid review

## Abstract

The coronavirus (COVID-19) pandemic struck globally and has affected higher education institutions (HEIs) and their operations, indirectly impacting the progress of the Sustainable Development Goal 4 achieved thus far. This article addresses HEIs achievements and challenges experienced in the wake of the pandemic. Online news media reports played a facilitative role in providing information to the HEI communities. A rapid review exploring online news media messages relating to higher education at the onset of the COVID-19 pandemic in Africa was utilised. Narrative synthesis was used to analyse the data. The results highlight HEIs achievements, which aim to ensure that all students receive the same level of education and provision in terms of devices and mental health support. However, challenges were also experienced at HEIs and include students feeling uncertainty and fear regarding completing their education. Furthermore, the results also show that not all students received the same level of education due to contextual factors, thus deepening the existing social disparities in Africa. The pandemic provides an opportunity for HEIs to embed the components of global citizenship education into the curriculum and to work in an innovative way to promote Sustainable Development Goal 4.

## 1. Introduction

The coronavirus (COVID-19) health pandemic was, and is, an occurrence for which no one was prepared. It was first detected in Wuhan China, December 2019, and swiftly spread across the world. In its wake, it created a destructive path and foregrounded a devastating nexus of health, economic, political, cultural, and social consequences for humanity and society [1]. This nexus struck globally across and within societies, but it was especially destructive amongst the less privileged, and highlighted the resource-constrained factors of the poor [1,2]. These factors included, amongst others, fragile health systems, struggling economies, lack of or limited access to remote learning, teaching, and working, high rates of unemployment, poverty, and food insecurity [1,3,4]. According to UNESCO, higher education institutions (HEIs)and schools from 185 countries were closed, affecting 9.8 million African students attending HEIs [5]. The International Association of Universities conducted a global impact survey which included responses from Africa, America, Asia and Pacific, and Europe. It was reported that the African region had the highest percentage of HEIs with closed campuses due to COVID-19 compared to the other three regions. In terms of HEIs moving to online teaching and learning, it was reported that Africa was the only region where teaching was suspended or cancelled for a brief period before moving to online teaching and learning. However, only 29% of African HEIs were able to quickly transition to online teaching and learning compared to 85% of HEIs in Europe, 72% in America and 60% in Asia and Pacific regions. With the introduction of new-age teaching and learning for HEIs that were not prepared for the online transition, many challenges either resurfaced or became more evident [6]. This became evident as only 24% of the African population has internet access or experiences poor to no connectivity in certain rural areas [5], threatening the goal of inclusivity of learning approaches for students in higher education. Therefore, the greatest threat of this nexus, specifically within the African region, was the regression of development towards achieving the Sustainable Development Goals (SDG) [7].

SDG 4, which refers to education, plays a key role in development. SDG 4 intends to “ensure inclusive and equitable quality education and promote lifelong learning opportunities for all” [7] (p. 18). Furthermore, the 10 targets for SDG 4 stipulate that by 2030, “all learners should have equal access, be included, attend safe schools, increase scholarships for higher education, education should be lifelong and there should be an increase in the number of people with relevant skills, numeracy and literacy”. Specifically, Target 4.7 intends for learners to,

“…acquire the knowledge and skills needed to promote sustainable development, including, among others, through education for sustainable development and sustainable lifestyles, human rights, gender equality, promotions of a culture of peace and nonviolence, global citizenship and appreciation of cultural diversity and of culture’s contribution to sustainable development” [7].(p. 21)

Africa, as a continent, suffers from corruption, exploitation of power, unethical conduct and widening inequality gaps [8,9]. The Southern African Development Community (SADC), created with the idea to foster regional integration of the 14 member states (Angola, Botswana, Congo (DR) Lesotho, Malawi, Mauritius, Mozambique, Namibia, Seychelles, South Africa, Swaziland, Tanzania, Zambia and Zimbabwe) further struggles to forge gender equality, women’s economic empowerment, economic rights and growth, and financial security in the region [10,11]. SADC aims to achieve sustainable development, alleviate poverty and improve the living conditions, quality of life and wellbeing of its people [11]. One of the means of achieving these aims is by providing opportunities to engage in and acquire an education [7]. With the spread of the COVID-19 pandemic, the deprivation effects on various systems in society are of concern, specifically affecting the South African education system. Thus, resulting in subsequent effects on learners and students as the education system temporarily came to a standstill before adapting to online learning to continue through the pandemic. The COVID-19 pandemic resulted in many social restrictions including remaining indoors during the lockdown period to decrease the chances of people being exposed to this highly contagious disease. In South Africa, on 23 March 2020, the national lockdown alert level 5 was announced, and the implementation of lockdown was applied by the South African government on 26 March 2020, thus preventing students from physically attending all education institutions. On 1 May 2020, the South African president announced a gradual easing of the lockdown regulations to alert level 4, allowing only final year students in clinical training programmes to receive face-to-face tuition. By 1 June 2020, under alert level 3, HEIs were permitted to allow up to 33% of students from all years of study registered for clinical training programmes, as well as postgraduate students that require access to laboratories, to return to campuses [12,13]. By 17 August 2020, the announcement of alert level 2 allowed some HEIs the choice to welcome back 66% of their students to campus [13]. Some HEIs opted to allow students to return while other universities ruled out contact classes for the remainder of the 2020 academic year [13,14]. This resulted in students having to engage in online and blended learning within a context of constrained resources. While this became the only viable solution during the COVID-19 pandemic, many students at African HEIs did not have access to internet, experienced connectivity challenges, could not afford the costs of a device and data to connect to the internet [5,15]. Although, many universities partnered with internet providers to negotiate zero-rated access to specific educational learning management systems and information websites, the digital divide became more apparent. In addition, many academics with teaching responsibilities had little to no experience in the delivery of online teaching and had to quickly familiarise themselves with the various systems to ensure that online teaching continued [16]. Not all university staff owned laptops or personal computers, nor had proper access to the internet at home. In some cases, the laptops needed to be shared with a spouse or children who were being home-schooled [16]. Furthermore, meeting research outcomes associated with supervising postgraduate students or academics reaching their research projected outcomes were affected too [16,17].

The acquisition of knowledge and skills towards global citizenship education (GCE) is a key role player in the sustainability of societies. GCE includes respect for diversity, solidarity, and a shared sense of humanity [15]. This means that GCE is located within a human rights framework and encourages non-violence, social cohesion, inclusivity, social justice, tolerance, and democracy in developing cohesive and sustainable societies [15,18], while encouraging learners to be globally competent, socially responsible, and responsive and civically engaged [19]. These aspects of GCE are applied covertly and overtly in the acquisition of knowledge and skills in higher education Institutions (HEIs). In fact, there is an expectation that HEIs will embed GCE within the curricula as the intention is to contribute towards and engage in the social good of society [10,15,20,21] thus ensuring social responsibility, social responsiveness, and civic responsibility. Owing to the COVID-19 pandemic, the intended progress towards achieving the Sustainable Development Goals (SDGs) was threatened, particularly for education and higher education in Africa [22].

While science was catching up with the effects of the pandemic on society, online news media was ahead in raising awareness and advocating about the issues regarding the pandemic, monitoring disease management and behaviour of people [23]. Online news media also played a facilitative role in providing information to the HEI communities. How this was done is not very clear. This research study seeks to address the following question: What is the media focus in their description the impact of COVID-19 on stakeholders in higher education? This review, as part of an #OpenUpYourThinking research challenge hosted by the Joint Education Trust (JET) Education Services and sponsored by the United Nations Educational Scientific and Cultural Organisation (UNESCO), sought to explore the use of news report online media by HEI to promote key GCE factors during the pandemic in Africa through a rapid review. The objectives of the rapid review study included: (1) To identify the good factors projected in media during the COVID-19 pandemic regarding HEIs; (2) To identify the negative factors/challenges projected during the COVID-19 pandemic regarding HEIs; and (3) To identify the information presented and promoted by HEIs about COVID-19 in terms of curriculum, teaching and learning, research, and management.

## 2. Materials and Methods

A rapid review design and approach was utilised to explore the news reported online focussing on media messages relating to higher education at the onset of the COVID-19 pandemic in Africa. “Rapid reviews are a form of knowledge synthesis in which components of the systematic review process are simplified or omitted to produce information in a timely manner” [24]. Furthermore, systematic reviews are a type of literature review that uses systematic methods to collect secondary data, critically appraise research studies, and synthesize findings qualitatively or quantitatively [25].

A rapid review approach was used for two reasons, namely: (i) To synthesize the literature within a limited timeframe [26] and (ii) to support novice researchers through the rapid review training process conducted by JET Education Services and UNESCO. This study was conducted according to the guidelines of the Declaration of Helsinki and approved on 8 June 2020 by the Institutional Review Board (or Ethics Committee) namely Humanities and Social Research Ethics Committee of University of the Western Cape (protocol code HS20/5/2).

### 2.1. Search Strategy

Searches were limited to African websites including DispatchLive, eNCA, Daily Maverick, Inside Education, News24, Mail & Guardian, Times Higher Education, the Sunday Times Live, Bhekisisa, ADEA, University World News Global, IOL, The Conversation and Google News. Limiters were applied consistently to all searchers. These limiters included media published between March 2020 to June 2020. The selected time-period was chosen for the following reasons: HEIs shut down their normal operations from March 2020; it formed part of the #OpenUpYourThinking research challenge for novice researchers for a 4-month period; and to ensure the inclusion of real-time evidence, pertaining to the topic under review.

Forty search terms were used related to higher education institutions, universities, COVID-19, impact and stakeholders (staff, students, academics). The search focused on universities in Africa. The eligible online news articles uniform resource locator (URL) site was captured on an excel spreadsheet and duplicates were removed.

### 2.2. Inclusion and Exclusion Criteria

The online news URL website was recorded if they reported activities and changes in and for higher education and higher education stakeholders brought about by the COVID-19 pandemic. News reports were included if they were published in Africa and in English, between March 2020 and June 2020.

### 2.3. Screening and Eligibility

Teams of three or four researchers conducted the searchers. The teams included a lead researcher and two researchers from two African universities. This formed part of the research challenge hosted by JET Educational services and UNESCO. Each team was allocated four to five online news media platforms. The researchers on each team worked independently to search for the eligible news articles. This was then assessed by the remaining researchers on the team. Disagreements were resolved via discussions through online meetings to reach consensus. If consensus could not be reached, a third researcher was consulted.

### 2.4. Quality Assessment and Data Extraction

Data from 51 online media articles were independently extracted using a data extraction form adapted and developed from Roman and Frantz [27] to include the media platform, link, date extracted, reference, title, good/positive factors, negative factors, preparedness, responsiveness, change recommendations, vulnerability, support provided, leadership and governance, policy, the new normal regarding what is to be expected after lockdown, teaching and learning, and research [27]. The research teams identified the HE achievements during the COVID-19 pandemic as good and advantageous factors contributing toward HEIs processes as well the needs and well-being of the student and staff. The challenges in HE during the COVID-19 pandemic were identified as negative factors resulting in disadvantage points for HEI process, staff, and students alike. The results from all the news media platforms were collated into one excel spreadsheet, analysed for common themes using narrative synthesis and reported via a narrative summary.

## 3. Results

Although African websites were included, the results produced pertained to South African HEIs only. The narrative synthesis yielded two main aspects that have been narratively summarised and presented: (a) Higher education achievements during the COVID-19 pandemic; and (b) Challenges in higher education during COVID-19 pandemic (Table 1).

### 3.1. Higher Education Achievements during COVID-19

#### 3.1.1. University Processes–“No Students Left Behind”

After the news of the pandemic in Africa, South African HEIs made decisions regarding their operational processes (i.e., attending face-to-face classes and using the onsite support services) as well as contingency plans to ensure the safety of students and all staff members. Universities adopted a “no student left behind” approach with the global aim of saving lives amid the COVID-19 pandemic. As a start, HEIs contingency plans included starting the recess period earlier than planned to decrease the possible infection and spread of the coronavirus. In addition, the deregistration deadline was extended to prevent any untoward financial consequences for students who decide to deregister.


*Head of Universities SA Prof Ahmed Bawa told TimesLIVE that the underlying principle is that not a single student in the system should be left behind.*

*(2020-04-20-no-student-will-be-left-behind-by-e-learning-universities-sa)*



*“This means that every university will restructure the academic year so as to ensure that every student has a fair and equal opportunity to complete the academic year, even if it means going into the first part of 2021,” said Bawa.*

*(2020-04-20-no-student-will-be-left-behind-by-e-learning-universities-sa)*


Due to the uncertainty of the pandemic, there was no ‘business-as-usual’. Some universities prepared for a longer lockdown period as opposed to returning to normal operational processes, based on lessons learned from countries in the Northern Hemisphere, thus creating an adjustment on the planned academic year. The South African Department of Higher Education and Training (DHET) informed the Parliament that a decision to restructure the 2020 academic year, by extending the 2020 academic year into the beginning of 2021 was made. The restructure of the 2020 academic year included extending all formative and summative assessment due dates, as the first order of concern was to keep all students and staff safe.


*“What we have recognised is that given the time that has already been lost, there has to be a reorganisation of the academic year, and it’s possible that the end of the academic year would extend into the beginning of 2021,” said Diane Parker, the deputy director-general responsible for university education.*

*(2020-04-21-academic-year-likely-to-spill-over-into-2021-higher-education-ministry)*


All South African HEIs prepared to transition to online teaching and e-learning. This means that all online teaching materials are needed to reach all students. The most crucial element all tertiary institutions needed to consider was the different types of students they had and the support they needed. These considerations were imperative when planning the online material for online teaching and e-learning.


*Nzimande said they were aware that it was not only NSFAS [referring to National Student Financial Aid Scheme] funded students who require these devices and while the government cannot commit for the so-called missing middle students, they have made institutions aware of this. He said they are working with all institutions on the reprioritisation of funds that could be used to buy such devices for students who need them.*

*(2020-05-15-government-wants-to-be-in-charge-of-university-laptops-procurement)*


#### 3.1.2. Support Strategies Provided

With the adopted approach of “no student left behind”, certain support strategies were implemented for students at HEIs, which included free mental health support services made available online, providing transport services for students living on student residences, who are financially unable to return home and providing accommodation during the national lockdown period for foreign national and international students.


*The University of KwaZulu-Natal will provide transport services for students who cannot afford to travel home after the closure of the institution due to the coronavirus outbreak.*

*(2020-03-20-ukzn-to-transport-res-students-home-amid-coronavirus-closure)*


With the transition to online teaching and e-learning, the provision of devices and laptops were made to students in need. This would be done in a strategic manner to ensure a fair and just distribution to all students. With this, partnerships with data service providers were initiated to provide zero-rated data rates to all students.


*The department of higher education & training on Saturday announced that they have successfully negotiated favourable rates with all mobile network operators for National Student Financial Aid Scheme (NSFAS) beneficiaries and the Funza Lushaka students.*

*(2020-05-24-department-of-higher-education-secures-lower-mobile-rates-for-nsfas-beneficiaries)*


Students receiving bursaries from the South African bursary schemes, namely National Student Financial Aid Scheme (NSFAS) and Funza Lushaka, would continue to receive their funds even during the lockdown period. Students receiving NSFAS bursaries were given two additional years to complete the programme they are registered for. This was done to accommodate for certain factors, such as a death in the family or other socio-economic challenges and to prevent any further disadvantages to the students.


*The National Student Financial Aid Scheme (NSFAS) will continue to fund students while the 2020 academic year is underway, Higher Education Minister Blade Nzimande said in a briefing on Thursday. “Funding for all students will continue while the academic year is underway even during the lockdown,” he added.*

*(nsfas-funding-to-continue-while-academic-year-is-underway-nzimande-20200430)*



*MPs heard that the National Student Financial Aid Scheme (NSFAS) continues to pay students their allowances during this period, but that it recognised that with the extension of the academic year there may be a need to support students for additional months, possibly into the first quarter of 2021.*

*(2020-04-21-academic-year-likely-to-spill-over-into-2021-higher-education-ministry)*


#### 3.1.3. Social Responsiveness: Putting Skills and Knowledge to Use

The university student experiences taught them to be more resilient, and it is said that they should use this time to reflect on how they are adapting and creating a new way of working amidst the pandemic. They need to embrace their role as leaders and lead their families, communities, and societies by taking charge of their mindset, their attitude, their way of thinking, look at the positive aspects, and support each other. This can be done by listening to leaders that have been able to adapt and were transparent in their process. A suggestion of steps to follow: Lead by example, create awareness and educate the people in your community, spread the message to the elders and chiefs living in remote and rural communities, stop stigmatisation and discrimination and find more ways to come back from this situation better than before *(opinion-covid-19-how-students-can-play-a-critical-role-during-the-pandemic-20200330)*. This encourages students to change their way of thinking and behaviour toward social responsibility, responsiveness and civic engagement.


*University students can play a critical role during these difficult times and can be a part of the greater conversation in leading change in how we tackle Covid-19. It is your time to lead your families, communities, and societies by taking charge of the situation. The key is to listen to our adaptive leaders who have thoroughly and persistently given directives and have been transparent throughout the process. Here’s how you can assist: Lead by example; Educate people in your community; Spread the message to elders in remote communities; Stop stigma and discrimination and think of ways to bounce back (use this time to reflect).*

*(opinion-covid-19-how-students-can-play-a-critical-role-during-the-pandemic-20200330)*


As the pandemic grew and more people were getting ill, the government hospitals were filled to capacity and hospital staff had limitations in coping with the pandemic. Hundreds of health science students volunteered their time and services to assist the medical staff at tertiary hospitals in Cape Town, South Africa.


*Hundreds of health science students have organised themselves into volunteering groups to assist medical staff at Tygerberg Hospital in Cape Town during the Covid-19 pandemic. From making cloth masks to screening people for the virus, they are putting whatever medical skills and knowledge they have to use.*

*(2020-05-25-a-call-to-action-medical-students-volunteer-to-step-out-of-the-the-classroom-into-the-coronavirus-pandemic)*


#### 3.1.4. Intersectoral Collaborations-“The Pandemic Offers the African University a Fresh Start”

The pandemic resulted in an opportunity for HEIs, the government, the private sector, and non-profit organisations to work jointly in searching for strategies to ensure national health and safety. This included portable hand sanitisers, the production of personal protective equipment (PPE) and a COVID-19 testing kit. The evidence of the collaborations and research in the fight against COVID-19 has become a chance for HEIs to reinforce their credibility in informing government decisions and contribute to building public trust.


*The COVID-19 pandemic should only be used as a stage for a ‘great leap’ forward. The pandemic offers the African university a fresh start.*

*(from-a-pre-colonial-to-a-covid-19-post-university)*



*But perhaps this pandemic has also created an opportunity for science and evidence to regain credibility in informing government decision and public trust and for universities to demonstrate respect for evidence…Universities are making progress in manufacturing personal protective equipment, developing new technologies for non-intensive care unit provision of oxygen to Covid-19 patients, finding methods of testing for the virus to reduce turnaround times, and various other scientific advances.*

*(2020-05-08-covid-19-an-opportunity-for-universities-to-regain-public-trust)*


### 3.2. Challenges in Higher Education during the COVID-19 Pandemic

#### 3.2.1. Crisis-Era and Student Uncertainty

At the beginning of the pandemic in Africa, all South African tertiary education institutions were informed to shut down normal operations starting from the 18th of March 2020, two weeks before the announcement of the national lockdown alert level. With the shutdown of normal operations, which included face-to-face teaching and learning, the academic year needed to be saved. The Minister of Education responded with online teaching and e-learning as the best solution amidst the crisis. However, only 20% of Technical and Vocational Education and Training (TVET) colleges could afford to provide online education. This implied that not all HEIs implemented the e-learning multi-modals from the same time, thus leaving students behind and bringing social inequalities to the surface.


*The transition to online teaching and learning during the lockdown will lead to historically disadvantaged institutions and students from poor socio-economic backgrounds being left out.*

*(2020-05-18-poorer-institutions-limited-access-to-tech-platforms-worries-lecturers)*



*The universities that claim full readiness to switch to e-learning are simply engaged in a narrow and self-serving public relations exercise. Many poor black students, regardless of institutional affiliation, live in already congested homes which do not provide an enabling environment for e-learning. The signal strength of the network in some parts of South Africa is too weak to allow for uninterrupted e-learning.*

*(2020-05-18-poorer-institutions-limited-access-to-tech-platforms-worries-lecturers)*


Students from certain universities that were historically classified as Bantu universities believed that there was no clear plan for the 2020 academic year (*2020-05-05-student-union-rejects-two-tier-plan-to-save-the-academic-year)*. Students had no faith in the implementation of online teaching and learning, resulting in a significant decrease in students’ effort to perform academically. This further exacerbated the students’ uncertainty regarding their academic careers. The South African Union of Students rejected the plan to save the academic year via online teaching and learning, as this plan supported the HEIs that were resourced and not the resource constrained HEIs. Thus, extra support was needed.


*The South African Union of Students rejected the plan to save the academic year of the Minister of Higher Education as it supported advantaged universities and not disadvantaged universities.*

*(2020-05-05-student-union-rejects-two-tier-plan-to-save-the-academic-year)*


The pandemic resulted in additional costs in education for the government. The additional costs included training and development of staff, assistance in converting teaching content to the learning management systems, the procurement of data and digital devices for staff and students, and the payment of increased access to e-textbooks and copyright clearance. *(should-varsities-receive-a-covid-19-stimulus-package-20200423)*. This resulted in an unplanned increase in education costs and time required for the HEIs, staff and students, thus adding to the existing feelings of uncertainty creating by the COVID-19 pandemic. This was done to ensure that all students have access to education.


*Hence, when introducing emergency remote teaching and learning, the switch to a different pedagogy and approach, university management did not have sufficient time to restructure the fixed cost of the budget.*

*(should-varsities-receive-a-covid-19-stimulus-package-20200423)*



*The truth is Covid-19 has disrupted the lives and plans of not only students, but everyone else. The fact that things remain uncertain makes it even worse, causing anxiety about what is going to happen next. This brings another challenge, which is psychological. Most students are stressing about their future—the health effects of this are, of course, undesirable.*

*(2020-04-09-covid-19-and-south-african-universities-a-raft-of-problems-to-ponder)*


#### 3.2.2. Without the Provision of Support, Students Are Concerned They Will Not Succeed

In addition to the uncertainty pertaining to the academic year, students needed extra support to transition to online teaching and e-learning. Students have indicated that their greatest challenge is having to manage their e-learning workload. Peer learning is an important part of students’ learning process in terms of being able to study with their peers, ask questions if they are uncertain and to feel supported through their learning process. This was not possible during the pandemic, resulting in students feeling more stressed and anxious.


*One of the disadvantages of online learning is that you are not surrounded by people who are studying your course, so you can’t really engage with each other about the work.*

*(2020-05-25-dark-divide-the-very-different-experiences-of-students-trying-to-e-learn)*



*We haven’t started yet. It’s something that they’ve been talking about, and they keep sending out communiques to say they are still making plans.*

*(2020-05-25-dark-divide-the-very-different-experiences-of-students-trying-to-e-learn)*


Furthermore, students who qualify for and are recipients of bursaries from NSFAS were concerned whether they would still be receiving the awarded money. In addition, the ‘missing middle’ students would still not be supported by NSFAS. Furthermore, it was noted that 30% of the student body at one of the previously disadvantaged universities, do not have laptops and are unable to afford data to continue their online learning.


*Nzimande said while they acknowledge that students who fall within the so-called “missing middle” category, as well as students in private institutions also needed support in accessing data for their online learning, as many of them also come from homes that are hit hardest by the impact of the COVID-19 pandemic, unfortunately government is currently not in a position to subsidize them.*

*(2020-05-23-department-of-higher-education-secures-lower-mobile-rates-for-nsfas-beneficiaries)*



*...30% of the varsity’s 24,000-strong student body does not have access to devices like laptops, or even to data, while on lockdown at home.*

*(watch-uwc-appeals-for-public-help-so-no-student-will-be-left-behind-46740100)*


Certain students’ socio-economic contexts are not conducive to online learning as certain households may lack electricity, internet connectivity or physical space, thus the pandemic has exposed high levels of inequality leaving certain students behind highlighting the social disparities. In addition, students with disabilities, such as hearing-impaired students, needed to be taken in account, by ensuring that the online content had clear audio sound and captions and subtitles.


*“There is no plan for fringe groups, no plan for postgraduate students, no plan for students that must be on-site to fulfil their academic duties, as well as international students,” he said. “There are students with disabilities who are assisted by student support structures only available on campus to ensure that they navigate the demands of university, [and] students with psychological issues emanating from the demands of university or their domestic home situations, where they are now bound.”.*

*(2020-04-20-no-student-will-be-left-behind-by-e-learning-universities-sa)*


Students living in residences were told to evacuate the rooms immediately. This was of great concern as many of the students, including the international students, had no alternative accommodation. The international students were also fearful of returning home to prevent spreading the virus.


*Thus, the other challenge relates to the stranded international students who cannot travel back to their home countries at this critical time due to lockdown restrictions on international travel.*

*(2020-04-09-covid-19-and-south-african-universities-a-raft-of-problems-to-ponder)*


#### 3.2.3. Delays in Research and Postgraduate Student Projects

The pandemic also had a negative effect on research and postgraduate research projects. Many research projects were suspended or terminated due to the national lockdown regulations. This included experimental research projects which affected the achievement of possible medical solutions, and research projects which included animal studies as the animals could not be kept alive or maintained. The postponement of postgraduate research studies being conducted meant that their completion and graduation could be delayed by a year.


*The coronavirus outbreak has taken a heavy human toll and temporarily slowed malaria drug development by freezing new clinical trials and delaying existing ones. But as a reminder of how a pandemic can wreck lives and livelihoods, it reinforces the importance of infectious disease research.*

*(africa-must-have-research-and-treatment-tailored-to-its-reality)*



*Enrolling in university is a big achievement for many, especially those who are the first in their families to study towards obtaining a degree. But regulations around the containment of Covid-19 and the resulting changes may leave many students feeling as if they won’t achieve their goals of one day graduating varsity.*

*(disadvantaged-university-students-on-online-learning-some-dont-even-own-smartphones-20200403-2)*


## 4. Discussion

The results of the study revealed HEI achievements and challenges in higher education during the COVID-19 pandemic. The achievements identified by online news media reports revealed that HEIs took the stance to ensure that no students were left behind in terms of receiving quality education, material, financial and mental support; students engaged in community efforts by providing assistance to hospitals and medical staff; and opportunities arose to engage in intersectoral collaborations to strengthen partnerships between government, the private sector, and non-profit organisations. Conversely, the results showed that students were uncertain of their academic future; additional costs and expenses were added to HEIs, staff and students to transition to online teaching and learning; and research and postgraduate student projects came to a brief standstill, and students were concerned that this would delay their completion process. At the time of writing this manuscript, the traditional university in the SADC region continues to operate in crisis response mode to the COVID-19 pandemic. The rapid review approach used to collect data on news reported online messages pertaining to HEIs, proved valuable during the COVID-19 pandemic when HEI students and staff were uncertain about the future of academia. The lack of engagement in alternate ways of operating or functioning to promote access, modernisation and relevance significantly impacts on the ability of universities to respond to unanticipated challenges and contextual challenges that promote both citizenship and sustainable development. Crisis-era universities are dictated to by the prevailing socioeconomic and socio-political ideologies and landscapes. The opportunity to leapfrog into enhanced modes and levels of functioning in the face of phenomena like the pandemic is dependent on the extent to which the institutional culture promotes a culture of social competence, citizenry and civic engagement, social justice and responsiveness, innovation, excellence, and quality assurance within flexible governance [18], the GCE framework and the systems theory [28,29].

The systems theory allows for a holistic approach to the investigation of phenomena across multiple disciplines. The central concept, which dates back to Aristotle, is that the whole is greater than the sum of its parts [28]. In terms of HEIs, as social systems, there are sub-units that must interact in a harmonious manner for the organisation to be effective with integrated outcomes. As an ‘open system,’ interactions in the health, economic and political environments have an impact on the HEI system as a whole. Feedback loops allow for HEIs to adapt to changes in the environments. The ability to “integrate, build, and reconfigure internal and external competencies to address rapidly changing environments” is referred to as dynamic capabilities [30]. The fundamental dynamic capability categories are grouped into capabilities which are to (1) sense and shape opportunities and threats, (2) seize opportunities, and (3) maintain competitiveness by enhancing, combining, protecting, and, when necessary, reconfiguring the organisation’s intangible and tangible assets [30] (pg.1319). Strong dynamic capabilities aid an organisation’s innovation, success, and resilience, especially in the face of technological, economic and health uncertainty such as during the COVID-19 pandemic [31]. The dynamics of adaptation to change in HEIs becomes more prominent on a global platform.

In the document ‘Global Citizenship Education: An Emerging Perspective’ (2013), UNESCO refers to GCE as a frame of mind and a way of life. It speaks to the possibility of making a difference and making the world a better place for everyone [15,32,33]. The three components of GCE are global competence, social responsibility and responsiveness, and global civic engagement, and are subdivided and defined as follows: (i) Global competency refers to having an awareness and knowledge of the wider world, being cosmopolitan, actively seeking and trying to understand whilst respecting diverse cultures, and self-awareness of one’s limitations; (ii) Social responsibility and responsiveness is viewed as being an independent and critical thinker, possessing social concern for others, local communities and society. Being socially responsible and responsive means there is a recognition of global issues and social inequalities, possessing an altruistic and empathic nature and knowing that there’s an interrelation between local behaviour and global consequences; (iii) Global civic engagement refers to a demonstration of action, that is, responding to social, local and/global issues through community participation, volunteerism and/political and local activism in a practical or social and virtual media platforms to advance local and global programmes and agendas. The combination of the three GCE components lead to global citizenship [19,33,34]. Thus, GCE aims to develop cohesive and sustainable communities, through creating globally competent, socially responsible, and responsive and globally civic engaged students at HEIs [15,18,19,32]. The COVID-19 pandemic presented an opportunity for HEIs to integrate the components of GCE at their institutions, with staff and students alike.

Furthermore, to advance SDG 4, which is to “ensure inclusive and equitable quality education and promote lifelong learning opportunities for all” [7] (p.18), HEIs had to reconsider their stance on academic online learning delivery; their student support protocols; and support and services needed from external partnerships to enhance their operationalisation. As stated, HEIs are considered ‘open systems’ that adapt to the events within the economic, health and political environments. All sub-units within the HEI system, including but not limited to, students, staff, administrators, student and staff families, HEIs infrastructure, management and budgets, were considering how each sub unit would be affected [29]. To shift and adapt to online learning and teaching, a call for a re-engineering of the curriculum; development of guidelines for different and alternative modes of teaching and learning that specify the requirements for achieving learning outcomes, quality assurance and accreditation; and a large-scale investment in online education was made with urgency. Even though online learning has been used as part of distance learning for many years, there was an accelerated need to implement online learning across HEIs globally [35]. This initiative offered an opportunity to enrich learning materials with audio-visual materials and real-world examples to support theory and practice, also known as online learning. Online learning is a combination of elements that includes the internet, technology to convert learning materials to be accessible via the internet, delivery of instruction in a comprehensive manner and then the management of the programme and platform [36,37]. The two types of online learning, namely synchronous and asynchronous online learning platforms, were considered as a solution to the crisis that HEIs found themselves in. This was done keeping the object of “no student left behind” in mind, to ensure education for all [7], social responsiveness and justice, and a decrease in possible disparities that may arise. However, for online learning to be most effective, HEIs and lecturers should have an extensive understanding of how to operationalise the two types of platforms [6,38]. Thus, increasing the probability of lecturers’ new methods of teaching and instruction delivery [6]. In turn, this promoted SDG 4, that the acquisition of knowledge, skills and education be sustained throughout the throes of the pandemic. The recognition of the complexities involved in online teaching and learning included, but were not limited to, the provision of access to online learning material, student support, receiving and marking assignments and student assessments, connectivity, and access to devices, especially for students in resource-constrained environments. Therefore, the new reality of online teaching and learning calls for perpetual evaluation of the mode of delivery and the consequential quality implications of the didactic process. This new reality for lecturers and students created a social awareness and a deeper understanding of the diverse learning environments for students because of the pandemic. Thus, taking into consideration the inequalities and contextual realities of students and lecturers, providing strictly online teaching and learning may not be an option for students and lecturers in resource-constrained contexts [39,40]. This requires creative thinking in providing options to accommodate all students. Staff and students, including vulnerable students, required support to continue with the academic year and address their anxieties and uncertainty during the crisis-era [40].

As part of the HEIs ensuring that no student is left behind and academic operations for staff and students continue, support via external stakeholders was provided in terms of free mental health services, arranging for students to receive laptops and devices and students to continue receiving their bursary funds despite the HEIs being under lockdown. This was vital during the crisis as part of social responsiveness and to address any social disparities, as HEIs in Africa were unable to provide support on their own. As such, this created partnerships that would not have been forged until this pandemic crisis [41].

### New Information

The COVID-19 pandemic has changed many areas of life, including the outlook of life and behaviours, specifically how socialising will occur post-COVID-19. To date, many African universities have opened attempted to resume in person classes, but have not been successful due to COVID-19 reinfections occurring every four to five months. Within HEIs, the pandemic and its effects have highlighted the vast inequalities present [39]. Many HEIs and students have not had the resources to make a swift and comprehensive shift to online teaching and learning, thus impeding on the progress made to achieve SDG 4 by 2030 [7]. The recognition of a digital divide advocates the need for a more inclusive approach when considering vulnerable students, especially those with disabilities and mental health challenges. Furthermore, before in person class resumes and lecturers can return to the classroom, a redefinition of workplace practices and policies for flexible work arrangements that are less dependent on face-to-face time and include online learning are not only needed but necessitated. An example of this practice is hybrid teaching and learning which includes in person as well as online teaching and learning. There should be a change in the weighting of the assessments by creating smaller and more frequent assessments to ensure that students have more opportunity to learn and achieve favourable grades. Additionally, the governments should allocate funding to employ more qualified teaching tutors to assist lecturers with the teaching and grading workload. There should be a focus on well-defined and quality work outputs by changing task due dates and how tasks are executed as opposed to time management. HEIs will have to operate differently post-COVID-19 than before the pandemic (‘new normal’) [42]. This may seem like a daunting and challenging feat [43], but HEIs will continue, adapt, and grow [42]. The disruption of learning and daily living forces institutions to align with and adapt to the requirements to survive the Fourth Industrial Revolution. Virtual modes of distance learning have been required, as we prepare for prolonged disruption in the way education is delivered. The content of what is being taught, and the skills students are equipped with need to be embedded in the GCE framework in this ever-changing world [32,44]. While the advancement of technology and its use can deepen inequalities, the provision of digital literacies and skills to the student population facilitates the reduction of social inequalities [45]. Sharing of best practice and promoting impactful teaching, learning and research should be an imperative to advance the higher education agenda in Africa.

## 5. Conclusions

As HEIs in the Africa navigate their way through the pandemic, whilst keeping abreast of methods and operationalisation used at HEIs globally, the goal to create and maintain sustainable practices for knowledge and education delivery, and safety of staff and students, remains the priority. From online new media reports, events at HEIs involving the staff, students and operations were made known, thus enlightening, and providing a different perspective on how all students are experiencing education. The social inequalities were deepened and a public awareness and opportunity to address the social inequalities were made. The achievements and challenges experienced have forced HEIs to adjust their methods and will have an impact on HEIs post COVID-19 pandemic. There is a need for HEIs to document how they have adapted and adjusted to ensure the continuation of academia for all stakeholders [6]. Moving forward, HEIs will continue striving toward maintaining and promoting SDG 4 and creating a culture of global citizenship. The pandemic created an opportunity, albeit not under good circumstances, for HEIs to re-evaluate education delivery methods, and to promote sustainable development through education, social justice and global citizenship [7].

## Figures and Tables

**Table 1 ijerph-18-12888-t001:** Results of study.

Higher Education Achievements during the COVID-19 Pandemic					
1. University processes– “No students left behind”	2. Support strategies provided		3. Social responsiveness: Putting skills and knowledge to use		4. Intersectoral collaboration–“The pandemic offers the African university a fresh start.”
Challenges in higher education during the COVID-19 pandemic					
1. Crisis–era and student uncertainty		2. Without the provision of support, students are concerned they will not succeed.		3. Delay in research and postgraduate student projects	

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
