# Peer review of "Achievements and Challenges for Higher Education during the COVID-19 Pandemic: A Rapid Review of Media in Africa"

_ijerph, 2021, doi:10.3390/ijerph182412888_

Round 1
Reviewer 1 Report
The topic is of interest and novelty.It would be interesting and I suggest introducing a state
of the existing bibliography-literature on the subject of study,
a hypothesis and defining the objectives well. Regarding the method,
describe the methodology and type used, describe the type of documentation,
sources, etc. And in the discussions, establish a more detailed methodological
relationship between literature, hypotheses, objectives and results.
Reviewer 2 Report
This article lacks innovation and does not make good reference to the relevant research results over the past year. There is no general concept or theory, just a description of the facts.
Reviewer 3 Report
Congratulations for an interesting paper on a really relevant topic in today’s world. The paper is clearly designed and it shows a deep, thorough analysis of both the literature and the data obtained.
In my opinion, some minor changes are required:
- No DOI present in your references. Please include DOI for papers whenever possible.
- I think more (recent) references would be welcome in the 2 last paragraphs of the introduction, to further sustain the statements made.
- I found no tables, charts or figures to show the results obtained in a more graphical way. I do encourage you to provide graphical representation of the information presented, as it makes it easier for the reader to fully comprehend the main points of the paper.
- In the Conclusion, some words are devoted to what should be done in the future according to the results obtained, but little is said as to how it should be achieved. Please expand.
- Finally, reference [1] is wrong in the References List.
Other than that, a really interesting paper.
Reviewer 4 Report
Dear authors, please see some comments below.
Abstract:
Lines 12-3: impacting the progress of the Sustainable Development Goal 4 has achieved thus far
Lines 20-22: Unite the 2 sentences: "Furthermore, the results also show that not all students received the same level of education due to contextual factors, thus deepening the existing social disparities in Africa."
Lines 23-4: to work in an innovation (innovative) way
Introduction:
Line 39: United Nations Sustainable Development Goals (insert reference when you first mention them)
Line 40: speaks to education (refers to education)
Lines 67-79: In this paragraph are you referring to the 14 member states of SADC? The paragraph is very general and it would be useful to know more specifically which states you are talking about and whether there are differences between them.
For example, you write: "The implementation of lockdown was applied by governments and thus prevented students from physically attending all education institutions." Where? For how long? Afterwards did students have access to online education? What are the differences between African states? Reference [15] mentions South African HEIs.
Materials and Methods:
Lines 113-4: "Published between March 2020 to June 2020. Articles were included if they were published in Africa and in English." (Articles were included if they were published in Africa and in English, between March 2020 and June 2020.)
Line 129: What do you mean by this phrase: "way forward out of lockdown"?
Results:
Lines 143-150: In this paragraph are talking about HEIs across Africa or only in South Africa? It is utterly important that you clarify this aspect throughout your article. For example, you continue for the rest of section 3.1 to refer only to South African HEIs (as far as I could understand).
Line 222: "A suggestion of steps to follow" (perhaps insert reference here)
Line 283: "Students from certain universities" (Such as? Give examples, insert references).
Lines 299-310: Long quote, not analysed enough (personal input from the authors)
Discussion
Line 411: Insert new paragraph starting with "Furthermore" (the text is read with difficulty)
References
Check reference [1]
Link for reference [4] and [12] is broken.
Round 2
Reviewer 2 Report
This article could be accept.
Reviewer 4 Report
Dear authors,
I have read your cover letter and reviewed your paper.
All the best.